# Investigation of the Optimum Display Luminance of an LCD Screen under Different Ambient Illuminances in the Evening

Ying Zhou [1,2,†], Haoyue Shi [3,†], Qing-Wei Chen [1,†], Taotao Ru [1,2,*] and Guofu Zhou [1,2]

1   Lab of Light and Physiopsychological Health, National Center for International Research on Green Optoelectronics, South China Normal University, Guangzhou 510006, China; zhouying@m.scnu.edu.cn (Y.Z.); qingwei.chen@m.scnu.edu.cn (Q.-W.C.); guofu.zhou@m.scnu.edu.cn (G.Z.)
2   Guangdong Provincial Key Laboratory of Optical Information Materials and Technology & Institute of Electronic Paper Displays, South China Academy of Advanced Optoelectronics, SCNU, Guangzhou 510006, China
3   Guangdong Justice Police Vocational College, Guangzhou 510000, China; shihaoyue94@126.com
*   Correspondence: taotao.ru@m.scnu.edu.cn
†   These authors contributed equally to the work.

**Abstract:** Ambient illuminance and screen luminance have a significant influence on the visual fatigue and visual performance associated with the use of computers. The current study was conducted to investigate optimal screen luminance under different ambient illuminances and fit a curve of the optimum luminance of LCD screens under evening illumination. Thirty-three participants were assigned to rate screen brightness, visual comfort with screen luminance, satisfaction with ambient illuminance and visual fatigue under six screen luminance levels (3.87, 21.47, 42.74, 64.12, 84.77 and 106.7 cd/m$^2$) combined with five ambient illuminance levels (0, 25, 50, 75 and 100 lx) in the evening. The results showed that optimum LCD screen luminance increased with increasing ambient illuminance. Moreover, ambient illuminance and screen luminance levels should be in the range of 13.08–62.16 lx and 20.63–75.15 cd/m$^2$, respectively, to obtain the optimal subjective feelings of visual fatigue and visual comfort during the evening.

**Keywords:** ambient illuminance; screen luminance; visual fatigue; visual comfort; computer vision syndrome; evening





## 1. Introduction

Computer vision syndrome (CVS), also referred to as digital eye strain, describes a group of eye and vision-related problems (i.e., visual discomfort, visual fatigue, blurred vision, eye strain, dry eyes, irritated eyes, double vision, vertigo/dizziness, polyopia, headaches, neck pain, and difficulty refocusing the eyes) that result from prolonged computer, tablet, e-reader and cell phone use [1–4]. Among these problems, visual discomfort and visual fatigue are core symptoms. The prevalence of CVS has been reported to range from 25% to 93% of computer users depending on the instrument employed, sample examined and methodology used [3]. Despite the varied prevalence of CVS, as many as 90% of digital device users experience symptoms of CVS [4], nearly 60 million people are estimated to suffer from CVS globally, and new cases will increase at a rate of 1 million each year [5]. Multiple factors contribute to the severity of CVS, which can roughly be classified into three clusters: eye-related, environment-related and device-related [4]. Environment- and device-related factors are easily self-determined by users and, thus, should be prioritized in intervention studies. Among all the influential factors, ambient illuminance (which is categorized as an environment-related factor) and screen luminance (which is categorized as a screen-related factor) seem to be the most influential [1,2] and have attracted much attention in previous studies; thus, they are the focus of the current study.

Visual discomfort and visual fatigue have been studied in the light and lighting fields during the last half century [6–8]. Light is not only a prerequisite for vision but also a

beneficial factor in improving visual health if used in the correct manner (see review [9]). Despite the extensive research performed on this topic, previous findings have been diverse, and it is difficult to obtain a consensus. For example, some studies have found that compared with low (200 lx) and high ambient illuminance (800 or 1000 lx), medium illuminance (500 or 600 lx) is most beneficial for improving visual performance [10,11], but other research has revealed no significant visual performance improvement with ambient illuminance [12,13]. Compared with high ambient illuminance (700 lx), low illuminance (200 lx) has been shown to be slightly more preferred and better for visual recognition performance [14]. Although subjective visual fatigue was shown to be higher under 200 lx than 500 lx in one study, the objective visual fatigue evaluated by critical flicker frequency (CFF, which is an objective indicator of visual fatigue) was not as severe under the 200 lx condition as that under the 500 lx condition [15]. Moreover, visual comfort remains stable when the illuminance level achieves a comfortable threshold [16], and high illuminance (i.e., 1000 lx) can even cause dissatisfaction and discomfort and increase visual fatigue [17,18]. All these results suggest that relatively low or medium levels of ambient illuminance may be beneficial for enhancing visual comfort and minimizing visual fatigue.

In addition to ambient illuminance, screen luminance is another vital factor influencing visual discomfort and visual fatigue. Studies investigating the effects of screen luminance on visual work efficiency have revealed that compared with low screen luminance, high screen luminance significantly improves visual display quality and task performance [19]. High screen luminance has been shown to visually reduce image distortion and improve image quality, which is beneficial for task performance [20]. However, several studies have shown that working under high screen luminance for a long period elicits more visual fatigue (see review [2]). Na and Suk (2015) investigated the optimal luminance under extremely low illuminance levels (less than 1 lx) and found that the optimal levels were 10 cd/m$^2$ for the initial viewing condition and 40 cd/m$^2$ for the continuous viewing condition [21].

Ambient illuminance and screen luminance are frequently examined separately in different studies, and only a few investigations have studied the impact of ambient illuminance and screen luminance levels on visual fatigue and visual performance. For example, Benedetto et al. (2014) explored the combined effects of these two factors on visual fatigue and performance by manipulating the ambient illuminance (5 and 85 lx) and screen luminance (20 and 140 cd/m$^2$) simultaneously between subjects. They found that although subjective visual fatigue was not affected by either screen luminance or ambient illuminance, objective visual fatigue evaluated by blink frequency increased under high luminance conditions (140 cd/m$^2$) [22]. Using limited levels of ambient illuminance and screen luminance, this line of research provides a relatively better solution for visual fatigue. However, with regard to practical settings, the levels of illuminance and luminance employed would vary significantly, and some levels may be easily neglected by this line of research. To solve this problem, multiple levels of illuminance and/or screen luminance, with a wide range, need to be manipulated to determine the optimal screen luminance under different ambient illuminances. The related studies are summarized in Figure 1. Li et al. (2013) investigated the optimum screen luminance of mobile phones (Samsung S5660, 3.2 inch TFT-LCD screen, 480×320 pixels) under ambient illuminance by measuring task performance, CFF and subjective ratings of comfort, visual fatigue and overall satisfaction. The results of these studies showed that optimum screen luminance under ambient illuminance levels of 0, 100 and 500 lx corresponded to 11, 68 and 257 cd/m$^2$, respectively [23]. Using the same mobile phone, Zhang et al. (2013) revealed that optimum screen luminance levels under outdoor illuminance levels of 10k, 20k and 50k lx were 354, 734 and 1375 cd/m$^2$, respectively [24]. In a study by Ye et al. (2014), participants were required to evaluate the display quality and visual fatigue associated with three different mobile phones (Samsung Galaxy S3, Nokia 920 and iPhone 4S) to investigate the optimal luminance under variable levels of ambient illuminance. They found that the optimal luminance levels for 0, 150, 300, 400, 600 and 800 lx were 55, 170, 308, 436, 470 and

496 cd/m$^2$, respectively [25]. Yu et al. (2018) found that the optimal luminance for short-term reading was 58.5, 66, and 84 cd/m$^2$ for 100, 150, and 300 lx, respectively [26]. Recently, Kim et al. (2017) identified the comfort zone of display luminance under different ambient illuminances [27]. In a follow-up study, Yu et al. extended the reading duration and found that the optimal luminance zones were 45–90 cd/m$^2$ under 100 lx and 45–270 cd/m$^2$ under 230 lx [28]. These findings indicated that optimum screen luminance increased with increasing ambient illuminance, which was in accordance with Weber–Fechner's Law of brightness perception [29,30]. More specifically, brightness perception, an attribute of visual perception in which a source appears to be radiating or reflecting light, depends on the relative weight between screen luminance and ambient illuminance. It is essential to increase the screen illuminance accordingly with the incensement of ambient illuminance to keep the brightness perception unchanged.

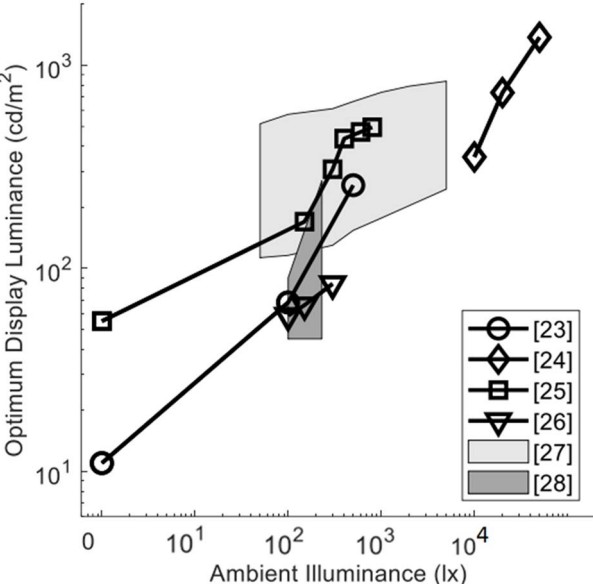

**Figure 1.** Previous findings on optimum display luminance under different ambient illuminance conditions for LCD screens.

Light brings multiple benefits to both individuals and society [31,32]; however, inappropriate light exposure (both ambient light and the light emitted from screens of digital devices), especially at night, can also potentially damage human biological functioning (i.e., circadian rhythm and nighttime sleep) [33–36]. This is the so-called nonvisual function of light, which has attracted much attention in the field of chronobiology and lighting. Considering the undesirable nonvisual effects of light on humans, both ambient illuminance and screen luminance are recommended to be manipulated at relatively low levels during the evening and night [37,38]. However, the effects of light on visual experiences and performance, in addition to biological function, should not be neglected. The lower illumination of screens or dimmer ambient illuminance may elicit many visual problems, such as lower legibility of reading and higher visual fatigue. The timings of the experiments administered were not reported in most of the previous studies regarding the combined effects of ambient illuminance and screen luminance on visual experience [23,25–28]. In addition, the optimum levels of screen luminance and ambient illuminance obtained during daytime (such as 1375 cd/m$^2$ [24]) would be too high for computer users during the evening [36,39]. In fact, the effects of light depend on the time of day [40,41]. The findings from diurnal studies may not be directly transferrable to nocturnal situations. Therefore, the optimal ambient illuminance and/or screen illumination levels for people using LCD computers at night are largely unknown.

The current study aimed to use subjective ratings to investigate the optimum display luminance of a computer LCD screen under different ambient illuminances and to plot a fitted curve of optimum screen luminance under different ambient illuminances in the evening.

## 2. Methods

### 2.1. Design

The study employed a 5 (Ambient Illuminance: 0, 25, 50, 75, and 100 lx) $\times$ 6 (Screen Luminance: 3.87, 21.47, 42.74, 64.12, 84.77 and 106.7 cd/m$^2$) within-subject design.

### 2.2. Participants

Thirty-three undergraduates (18 females and 15 males, mean age = 19.45 $\pm$ 1.82 years old) were recruited at a local university. All volunteers were screened and reported normal or corrected-to-normal vision, and none of them suffered from color blindness. The protocol was approved by the Institutional Review Board of the local university, and written informed consent was obtained from each participant before the start of the laboratory study.

### 2.3. Subjective Evaluations

#### 2.3.1. Screen Brightness

The screen brightness was evaluated by a 7-point Likert scale ranging from "1", being extremely dark, to "4", being neither too dark nor too bright, and "7", being extremely bright [42].

#### 2.3.2. Visual Comfort with Screen Luminance

Visual comfort with screen luminance was evaluated by a 5-point Likert scale ranging from "1", being very uncomfortable, to "3", being neutral, and "5", being very comfortable.

#### 2.3.3. Satisfaction with Ambient Illuminance

Satisfaction with ambient illuminance was rated on a 5-point Likert scale ranging from "1", being very unsatisfied, to "3", being neutral, and "5", being very satisfied.

#### 2.3.4. Visual Fatigue

Visual fatigue was assessed by the questionnaire retrieved from Chen et al. (2014) [43], which was adapted from Heuer et al. (1989) [44] and Iribarren et al. (2001) [45]. Seven items (eyestrain, eye ache, watery eyes, dry eyes, headache, blurred vision and continuous blinking) were used to evaluate the severity of visual fatigue by a 5-point Likert scale, with "1" representing "no such symptom" and "5" representing "extremely serious". The scores of seven items were averaged to indicate the severity of visual fatigue.

### 2.4. Experimental Setup

The experiment was conducted in a soundproof room (3.6 m $\times$ 3.6 m $\times$ 4.1 m) with an off-white wall (90.12% reflectance) and ceiling (90.12% reflectance) and dark gray floor (18.88% reflectance). Four workstations were separated by partitions. In each workstation, there was a light gray desk (1.2 m $\times$ 0.8 m, 50.77% reflectance) and one black chair. One white all-in-one computer (Lenovo C260, 19.5 in) with headphones, a keyboard and a mouse were placed on each desk. Moreover, in order to avoid the potential effect of natural light, no window was present in the room. The room temperature was kept at 26 °C by an air conditioner. For more details regarding the laboratory room, please refer to Ru et al. (2019) [46], Mao et al. (2018) [47] and Yang et al. (2019) [48].

Ambient illuminance levels (25, 50, 75 and 100 lx at eye level) were achieved by adjusting an intelligent desk LED lamp (Winstone-HSD9012A, Highstar) and were measured at eye level by a calibrated spectroradiometer (JETI Specbos1202, JETI Technische Instrumente GmbH, Jena, Germany) when the computer screen was turned off. The spectral power dis-

tributions of different illuminance conditions are displayed in Figure 2 [49,50]. Moreover, the spectrally weighted α-opic illuminance levels and daylight equivalent illuminance (EDI) levels for each illuminance condition are listed in Table 1.

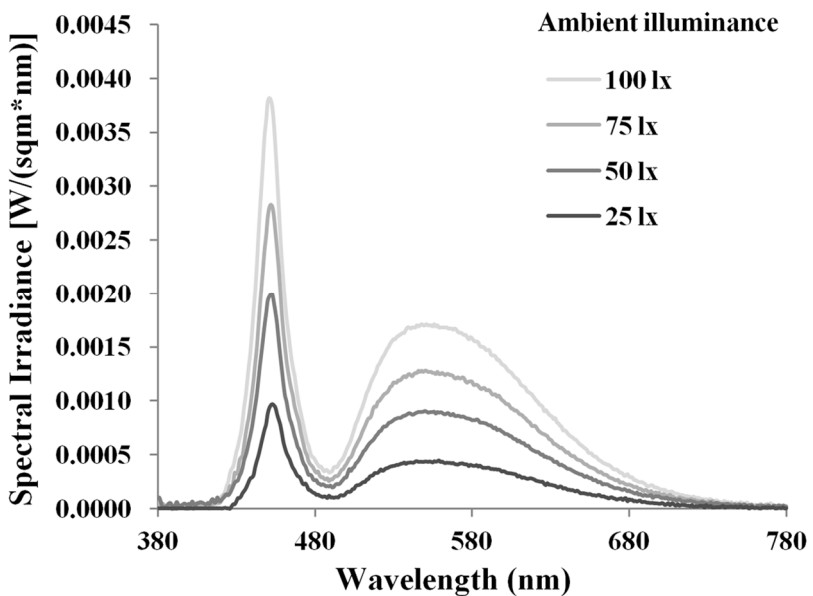

**Figure 2.** Spectral power distributions measured at eye level under 25, 50, 75 and 100 lx conditions.

**Table 1.** Spectrally weighted α-opic illuminance levels and daylight equivalent illuminance (EDI) levels under 25, 50, 75 and 100 lx conditions.

|  |  | α-Opic lx [49] | | | | α-Opic EDI [51] | | | |
|---|---|---|---|---|---|---|---|---|---|
|  | λmax (nm) | 25 lx | 50 lx | 75 lx | 100 lx | 25 lx | 50 lx | 75 lx | 100 lx |
| S-cone | 419.0 | 25 | 55 | 77 | 106 | 24 | 53 | 74 | 102 |
| Melanopsin | 480.0 | 22 | 46 | 65 | 86 | 20 | 42 | 58 | 78 |
| Rod | 496.3 | 23 | 49 | 68 | 91 | 21 | 44 | 62 | 83 |
| M-cone | 530.8 | 25 | 52 | 73 | 98 | 25 | 51 | 71 | 96 |
| L-cone | 558.4 | 25 | 51 | 72 | 96 | 25 | 52 | 74 | 99 |

One white all-in-one PC with a screen resolution of 1366 × 768 pixels was used in the current study. Screen luminance levels were measured by a JETI Specbos1202 at a distance of 60 cm from the computer screen when a pure white picture was displayed on the screen in dark ambient conditions. We employed six levels of screen luminance in the current study: 3.87 cd/m$^2$ (10%), 21.47 cd/m$^2$ (20%), 42.74 cd/m$^2$ (40%), 64.12 cd/m$^2$ (60%), 84.77 cd/m$^2$ (80%) and 106.7 cd/m$^2$ (100%).

The reading materials used in the formal laboratory study were six Chinese short novels. All reading materials selected in the current study were evaluated for likeability, understandability, and emotional arousal on a 5-point Likert scale by 10 other volunteers before the formal study. The results showed that the participants moderately liked the materials (*M* = 3.74, *SD* = 0.11), could understand the materials without difficulty (*M* = 3.12, *SD* = 0.05), and experienced slight affective arousal after reading the materials (*M* = 2.35, *SD* = 0.11). No significant difference in likeability (*F* (5, 24) = 0.23, *p* = 0.95), understandability (*F* (5, 24) = 0.35, *p* = 0.71), and emotional arousal (*F* (5, 24) = 0.12, *p* = 0.10) was revealed across the materials. Moreover, the reading materials were presented in black on a white background and uniformly arranged in 18-point Song font (character height was approximately 6 mm) with 1.5 times line spacing to maintain consistency.

## 2.5. Procedure

The formal experiment was conducted from 19:00 to 21:00. The participants were required to arrive at the laboratory before 19:00 and were instructed to sit at the workstation. The distance between the participants and the computer screen was kept at approximately 55–60 cm. The experiment started with a 2-min darkness adaptation period and consisted of six test sessions. Each test session contained five measurement blocks.

Following previous studies [27,42], the ambient illuminance increased sequentially from block 1 (0 lx) to block 5 (100 lx) because the speed of light adaptation is much faster than that of the dark adaptation in the human visual system [52], and the screen luminance was set randomly at one of six luminance levels for each session. In each measurement block, ambient illuminance lasted for 30 s, and the participants were asked to read the randomized presented novel on the computer screen. After each screen luminance and ambient illuminance combination, they were asked to evaluate screen brightness, visual comfort with screen luminance, satisfaction with ambient illuminance, and visual fatigue with questionnaires. The screen luminance and ambient illuminance remained unchanged until they finish all ratings during each measurement block. Between the two blocks, the participants had 1 min to rest, during which the participants were requested to close their eyes while the experimenters manipulated the ambient illuminance level. The order of the screen luminance was randomized between blocks. There was also 1 min between the sessions for participants to close their eyes to rest and for the experimenters to change the screen luminance. Multiple short breaks between blocks and test sessions were also used to relieve the potential stress imposed on the participants. An overview of the experimental procedure is shown in Figure 3, and the formal experiment lasted for approximately 76 min in total.

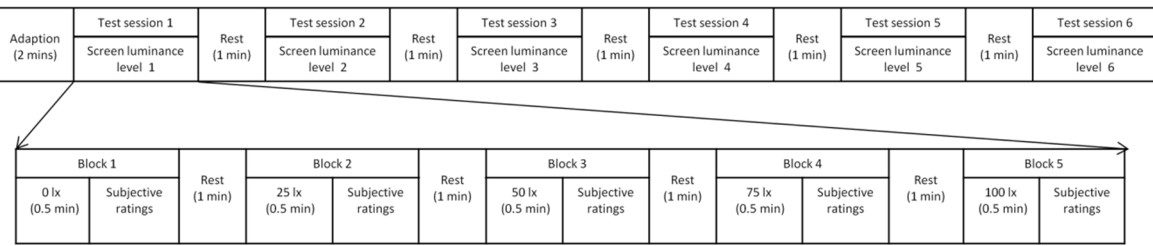

**Figure 3.** Overview of the experimental procedure.

## 2.6. Data Analysis

All analyses were performed using GraphPad Prism 8. First, the descriptive data (mean and standard error) on screen brightness, visual comfort with screen luminance, visual fatigue, and illuminance satisfaction were analyzed. A 5 (Ambient Illuminance: 0, 25, 50, 75, and 100 lx) × 6 (Screen Luminance: 3.87, 21.47, 42.74, 64.12, 84.77 and 106.7 cd/m$^2$) repeated analysis of variance (ANOVA) was applied to all subjective evaluations using SPSS version 21.0 (IBM, USA). All post-hoc tests were Bonferroni corrected for multiple comparisons. Note that all the results of ANOVA analysis were not included in the main text but could be found in the Supplementary Materials.

Second, the mean ratings on screen brightness, visual comfort with screen luminance, visual fatigue, and illuminance satisfaction were subjected to the "Nonlinear regression (curve fit)" module. Curve fitting has been used in previous studies on visual comfort [53,54] and brightness perception [55]. The general rule of curve fitting was to maximize adjusted R$^2$ without overfitting the model. "Polynomial" was selected, and "First order polynomial (straight line)", "Second order polynomial (quadratic)", "Third order polynomial (cubic)", "Fourth order polynomial", and "Fifth order polynomial" were tested sequentially. After each test, the curve fitting results were checked to determine if the model was overfitting. If the model was overfitting, the process of curve fitting was stopped, and the best function being tested was selected for curve fitting.

Third, after the determination of the best function, the curves were fitted for each dependent variable. For screen brightness, a rating of "4" means "neither too dark nor too bright", and, thus, a straight line across "4" was fitted to determine the optimum screen brightness. The intersection between this straight line and the best-fit straight line for each ambient illuminance condition was used to determine the optimum screen luminance for each ambient illuminance condition. For visual comfort with screen luminance, the max value for visual comfort was determined after the best-fit curve was plotted for each ambient illuminance, and the corresponding screen luminance was the best one for this specific ambient illuminance condition. The minimum value for visual fatigue was determined after the best-fit curve was plotted for each ambient illuminance, and the corresponding screen luminance was determined as the best one for this specific ambient illuminance condition to maintain visual fatigue at a relatively low level. The max value for illuminance satisfaction was determined after the best-fit curve was plotted for each screen luminance, and the corresponding ambient illuminance was determined as the best one for this specific screen luminance condition. To determine the best ambient illuminance for visual fatigue, the mean ratings for visual fatigue were modeled again based on ambient illuminance. The minimum value for visual fatigue was determined after the best-fit curve was plotted for each screen luminance, and the corresponding ambient illuminance was determined as the best one, for this specific screen luminance condition, to keep visual fatigue at a relatively low level.

After step 3, the optimum screen luminance for screen brightness, visual comfort and visual fatigue under different ambient illuminance conditions was determined, and was used for curve fitting. Moreover, the optimum ambient illuminance for satisfaction and visual fatigue under different screen luminances was also determined, and was utilized to fit the curve.

## 3. Results

### 3.1. Subjective Evaluations

#### 3.1.1. Screen Brightness

The descriptive data for screen brightness are shown in Table 2. The current data indicated that participants' perceived brightness increased with screen luminance but decreased with ambient illuminance.

**Table 2.** The descriptive data for screen brightness (M ± SE).

| | | Screen Luminance (cd/m²) | | | | | |
|---|---|---|---|---|---|---|---|
| | | **3.87** | **21.47** | **42.74** | **64.12** | **84.77** | **106.7** |
| | 0 | 2.85 ± 0.15 | 3.33 ± 0.18 | 4.39 ± 0.21 | 4.76 ± 0.24 | 5.48 ± 0.18 | 5.91 ± 0.15 |
| | 25 | 2.64 ± 0.18 | 3.21 ± 0.14 | 4.06 ± 0.14 | 4.67 ± 0.17 | 4.73 ± 0.16 | 5.12 ± 0.11 |
| Ambient illuminance (lx) | 50 | 2.45 ± 0.16 | 2.85 ± 0.12 | 4.09 ± 0.12 | 4.61 ± 0.18 | 4.76 ± 0.15 | 5.03 ± 0.12 |
| | 75 | 2.42 ± 0.17 | 2.76 ± 0.12 | 4.24 ± 0.19 | 4.55 ± 0.22 | 4.67 ± 0.12 | 4.97 ± 0.13 |
| | 100 | 2.33 ± 0.20 | 2.61 ± 0.19 | 4.09 ± 0.15 | 4.48 ± 0.15 | 4.64 ± 0.17 | 4.79 ± 0.16 |

#### 3.1.2. Visual Comfort with Screen Luminance

The descriptive data for visual comfort with screen luminance are shown in Table 3. Participants' subjective visual comfort was greatest at a screen luminance of approximately 42.74 cd/m² and ambient luminance between 25 and 50 lx.

**Table 3.** The descriptive data for visual comfort with screen luminance (M ± SE).

| | | Screen Luminance (cd/m²) | | | | | |
| --- | --- | --- | --- | --- | --- | --- | --- |
| | | 3.87 | 21.47 | 42.74 | 64.12 | 84.77 | 106.7 |
| | 0 | 2.67 ± 0.15 | 3.03 ± 0.13 | 2.82 ± 0.18 | 2.73 ± 0.15 | 2.21 ± 0.15 | 2.12 ± 0.12 |
| Ambient illuminance (lx) | 25 | 2.91 ± 0.13 | 3.18 ± 0.13 | 3.55 ± 0.09 | 3.03 ± 0.11 | 2.82 ± 0.13 | 2.88 ± 0.14 |
| | 50 | 2.79 ± 0.13 | 3.00 ± 0.13 | 3.73 ± 0.08 | 3.27 ± 0.13 | 3.21 ± 0.13 | 3.27 ± 0.09 |
| | 75 | 2.42 ± 0.13 | 2.79 ± 0.12 | 3.36 ± 0.10 | 3.09 ± 0.13 | 3.18 ± 0.10 | 3.09 ± 0.12 |
| | 100 | 2.42 ± 0.15 | 2.24 ± 0.12 | 3.09 ± 0.13 | 2.97 ± 0.14 | 3.00 ± 0.13 | 2.91 ± 0.12 |

### 3.1.3. Illuminance Satisfaction

The descriptive data for illuminance satisfaction are shown in Table 4. The current findings showed that participants' satisfaction with ambient illuminance was greatest at a screen luminance of approximately 42.74 cd/m² and ambient luminance between 25 and 50 lx.

**Table 4.** The descriptive data for illuminance satisfaction (M ± SE).

| | | Screen Luminance (cd/m²) | | | | | |
| --- | --- | --- | --- | --- | --- | --- | --- |
| | | 3.87 | 21.47 | 42.74 | 64.12 | 84.77 | 106.7 |
| | 0 | 2.79 ± 0.15 | 2.88 ± 0.17 | 2.97 ± 0.15 | 2.79 ± 0.16 | 2.58 ± 0.16 | 2.45 ± 0.14 |
| Ambient illuminance (lx) | 25 | 3.27 ± 0.10 | 3.27 ± 0.14 | 3.39 ± 0.12 | 3.18 ± 0.12 | 2.97 ± 0.13 | 3.00 ± 0.16 |
| | 50 | 2.91 ± 0.12 | 3.18 ± 0.13 | 3.58 ± 0.09 | 3.21 ± 0.10 | 3.33 ± 0.10 | 3.06 ± 0.12 |
| | 75 | 2.73 ± 0.17 | 3.00 ± 0.14 | 3.00 ± 0.14 | 2.64 ± 0.15 | 3.03 ± 0.15 | 2.82 ± 0.17 |
| | 100 | 2.33 ± 0.18 | 2.61 ± 0.14 | 2.58 ± 0.18 | 2.33 ± 0.17 | 2.79 ± 0.15 | 2.58 ± 0.16 |

### 3.1.4. Visual Fatigue

The descriptive data for visual fatigue are shown in Table 5. Participants' visual fatigue was lowest at a screen luminance between 21 and 42 cd/m² and at an ambient illuminance between 25 and 50 lx.

**Table 5.** The descriptive data for visual fatigue (M ± SE).

| | | Screen Luminance (cd/m²) | | | | | |
| --- | --- | --- | --- | --- | --- | --- | --- |
| | | 3.87 | 21.47 | 42.74 | 64.12 | 84.77 | 106.7 |
| | 0 | 2.18 ± 0.15 | 1.91 ± 0.13 | 2.12 ± 0.15 | 2.27 ± 0.15 | 2.48 ± 0.15 | 2.42 ± 0.16 |
| Ambient illuminance (lx) | 25 | 2.09 ± 0.13 | 1.76 ± 0.12 | 1.88 ± 0.13 | 2.18 ± 0.14 | 2.24 ± 0.12 | 2.09 ± 0.13 |
| | 50 | 2.18 ± 0.13 | 1.85 ± 0.12 | 1.82 ± 0.10 | 2.09 ± 0.13 | 1.85 ± 0.12 | 1.97 ± 0.12 |
| | 75 | 2.30 ± 0.15 | 1.91 ± 0.13 | 1.85 ± 0.12 | 2.33 ± 0.13 | 1.97 ± 0.12 | 2.09 ± 0.10 |
| | 100 | 2.39 ± 0.14 | 2.09 ± 0.10 | 1.97 ± 0.12 | 2.24 ± 0.15 | 2.09 ± 0.12 | 2.24 ± 0.10 |

*3.2. Curve-Fitting with Optimum Screen Luminance under Different Ambient Illuminance Conditions*
3.2.1. Optimum Screen Brightness Curve vs. High Visual Comfort Curve vs. Low Visual Fatigue Curve for Screen Luminance under Different Ambient Illuminance Conditions

The curve fitting results of screen brightness (second-order polynomial), visual comfort (third-order polynomial) and visual fatigue (third-order polynomial) are shown in Figures 4–6, respectively. Table 6 lists the optimum levels of screen luminance under different ambient illuminance conditions. The functional expressions of these three fitted curves are presented in Table 7.

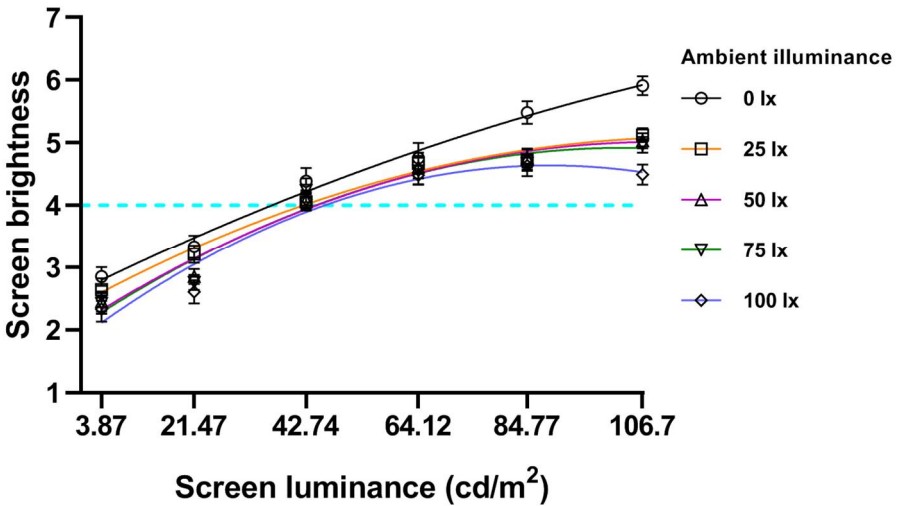

**Figure 4.** The curve fitting results of screen brightness.

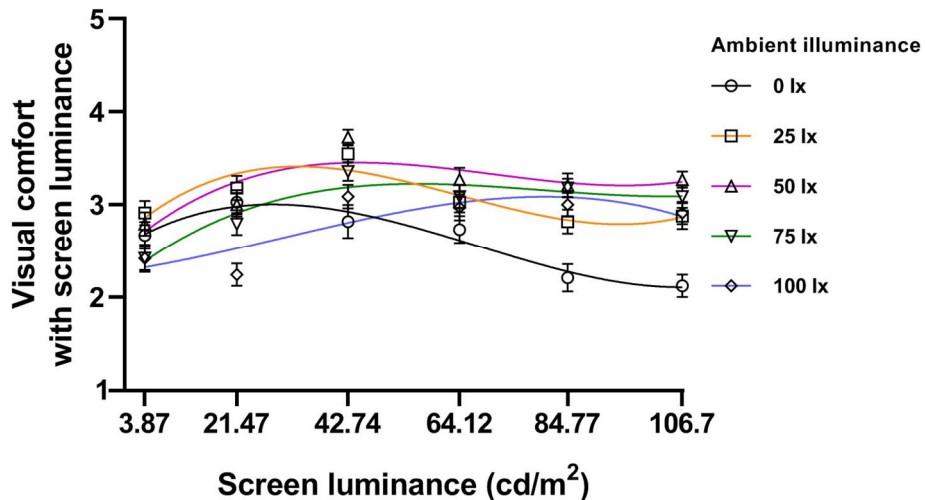

**Figure 5.** The curve fitting results of visual comfort against screen luminance.

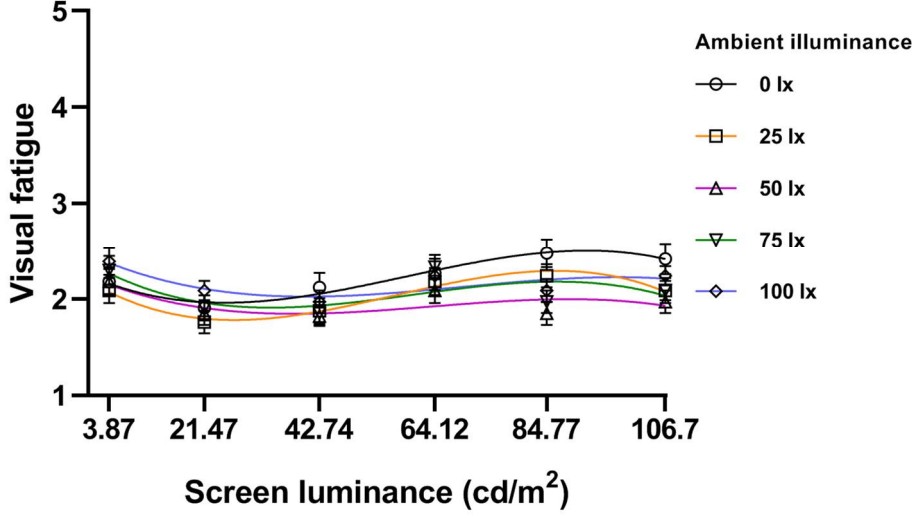

**Figure 6.** The curve fitting results of visual fatigue against screen luminance.

**Table 6.** The optimum screen luminance under different ambient illuminance conditions.

| Standard | Ambient Illuminance (lx) | | | | |
|---|---|---|---|---|---|
| | 0 | 25 | 50 | 75 | 100 |
| Optimum screen brightness | 36.20 | 42.21 | 44.74 | 44.57 | 46.24 |
| High visual comfort | 25.57 | 31.08 | 41.40 | 50.56 | 75.15 |
| Low visual fatigue | 20.63 | 23.01 | 34.50 | 28.47 | 35.59 |

The unit of screen luminance is cd/m$^2$.

**Table 7.** The fitted function of the optimum screen brightness curve, high visual comfort curve and low visual fatigue curve for screen luminance under different ambient illuminance conditions.

| Curve | Fitted Function | $R^2$ | *Adjusted* $R^2$ |
|---|---|---|---|
| Optimum screen brightness curve | 0.08976x + 38.3 | 0.8039 | 0.7385 |
| High visual comfort curve | 0.4746x + 21.02 | 0.9237 | 0.8982 |
| Low visual fatigue curve | 0.1415x + 21.36 | 0.7019 | 0.6026 |

$R^2$ represents the coefficient of determination.

The optimum screen brightness curve, high visual comfort curve and low visual fatigue curve for screen luminance under different ambient illuminance conditions are shown in Figure 7. The screen luminance levels for optimum screen brightness, high visual comfort and low visual fatigue all increased with increasing ambient illuminance levels. Regardless of the ambient illuminance level, the low visual fatigue curve was always below the two other curves. When the ambient illuminance changed from 0 lx to 100 lx, 20.63–75.15 cd/m$^2$ was the best range for screen luminance with respect to screen brightness, visual comfort and visual fatigue.

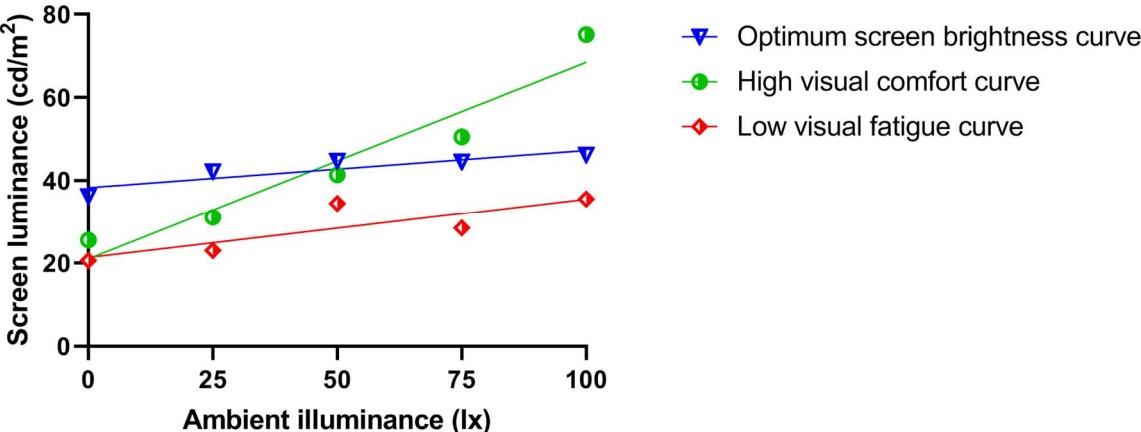

**Figure 7.** Optimum screen brightness curve, high visual comfort curve and low visual fatigue curve for screen luminance under different ambient illuminance conditions.

### 3.2.2. High Satisfaction Curve vs. Low Visual Fatigue Curve for Ambient Illuminance with Different Screen Luminance Conditions

The curve fitting results of illuminance satisfaction (second-order polynomial) and visual fatigue (second-order polynomial) are shown in Figures 8 and 9, respectively. Table 8 lists the optimum ambient illuminance under different screen luminance conditions. The functional expressions of these three fitted curves are presented in Table 9.

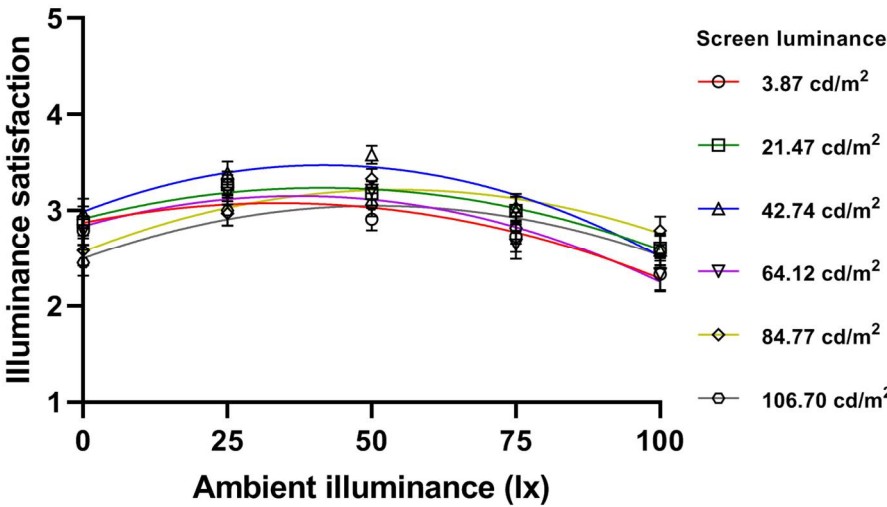

**Figure 8.** The curve fitting results of illuminance satisfaction.

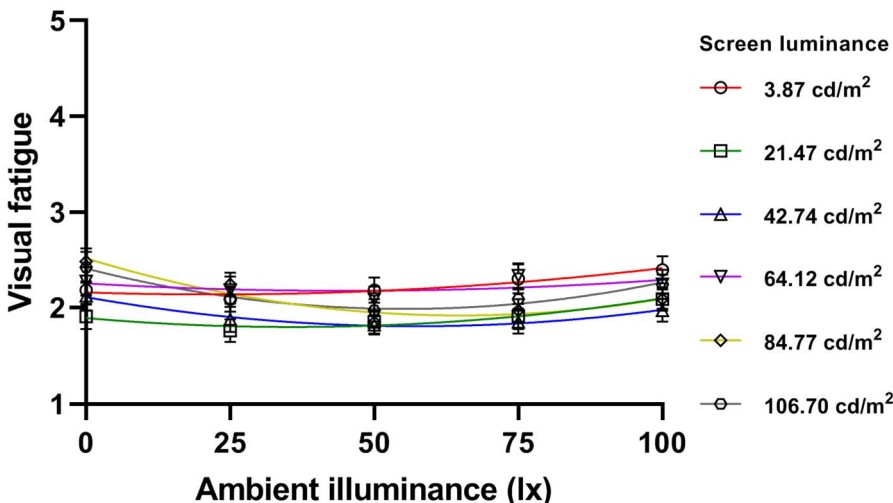

**Figure 9.** The curve fitting results of visual fatigue against ambient illuminance.

**Table 8.** The optimum ambient illuminance under different screen luminance conditions.

| Standard | Screen Luminance (cd/m$^2$) | | | | | |
|---|---|---|---|---|---|---|
| | **3.87** | **21.47** | **42.74** | **64.12** | **84.77** | **106.7** |
| High satisfaction | 26.80 | 35.05 | 38.08 | 33.59 | 48.16 | 48.95 |
| Low visual fatigue | 13.08 | 33.33 | 52.02 | 32.61 | 62.16 | 50.41 |

The unit of ambient illuminance is lx.

**Table 9.** The fitted function of a high satisfaction curve and low visual fatigue curve for ambient illuminance with different screen luminance conditions.

| Curve | Fitted Function | $R^2$ | Adjusted $R^2$ |
|---|---|---|---|
| High satisfaction curve | 2.617x + 18.01 | 0.798 | 0.7475 |
| Low visual fatigue curve | 3.378x + 7.72 | 0.5681 | 0.4602 |

$R^2$ represents the coefficient of determination.

The high satisfaction curve and low visual fatigue curve for ambient illuminance with different screen luminance conditions are presented in Figure 10. These results showed that the optimum ambient illuminance, in terms of satisfaction and visual fatigue, increased with increasing screen luminance. When the screen luminance changed from 3.87 cd/m$^2$ to 106.7 cd/m$^2$, the optimum ambient illuminance ranged from 13.08 lx to 62.16 lx.

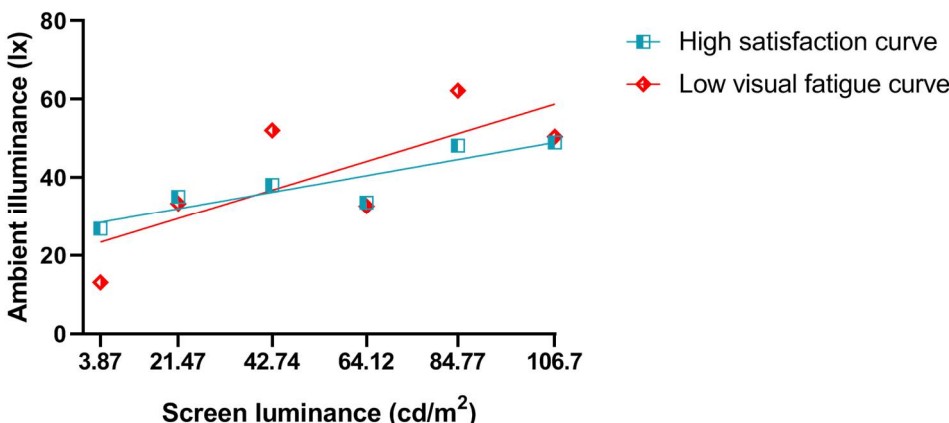

**Figure 10.** High satisfaction curve and low visual fatigue curve for ambient illuminance with different screen luminances.

Combining the results of Tables 6–9, and Figures 7 and 10, we recommended 13.08–62.16 lx and 20.63–75.15 cd/m$^2$ as the optimum levels for LCD screen viewing in the evening.

## 4. Discussion

Considering the widespread usage of LCD screens in the evening, the current study aimed to explore the optimum screen luminance for visual fatigue, visual comfort, satisfaction and brightness under varied ambient illuminance conditions at a relatively low level (ranging from 0 to 100 lx) for LCD screen users at night. Expanding on previous studies, the current study refined and employed five ambient illuminance levels and six screen luminance levels. The optimum screen luminance levels and the optimum ambient illuminance levels were determined and are reported in Tables 6 and 8, respectively. Moreover, the optimum screen brightness curve was fitted, and the high visual comfort curve and low visual fatigue curve for screen luminance, and the high satisfaction curve and low visual fatigue curve for ambient illuminance (see Figures 7 and 10), were evaluated based on subjective ratings of visual experience. To ensure that the requirements of visual comfort, visual fatigue, satisfaction and brightness were met, 13.08–62.16 lx and 20.63–75.15 cd/m$^2$ were shown to be optimal when using the LCD screen during the evening.

The results did show statistically significant effects of ambient illuminance and screen luminance on subjective fatigue and visual comfort. These findings were not in accordance with the study by Benedetto et al. (2014) [22]. The difference in the manipulations of ambient illuminance and screen luminance would partly explain these inconsistent findings. Only two ambient illuminance levels (5 and 85 lx) and two screen luminance levels (20 and 140 cd/m$^2$) were investigated in the study by Benedetto et al. (2014), while multiple levels of both ambient illuminance and screen luminance were employed in the current study, making it less likely to obtain significant main effects of ambient illuminance and screen luminance. Meanwhile, the current study employed subjective ratings of visual fatigue and visual comfort, while eye blink frequency was employed as an objective indicator of visual fatigue in their study.

In addition to the influence of ambient illuminance and screen luminance on visual fatigue, the current findings revealed that the optimum screen luminance increased accordingly with increasing ambient illuminance levels, which was in accordance with most previous studies [23–28]. For example, Li et al. (2013) found that the optimum screen luminance of a mobile phone was 11 cd/m$^2$ and 68 cd/m$^2$ under 0 lx and 100 lx conditions,

respectively [23]. Ye et al. reported that the optimal luminance level was 55 cd/m$^2$ and 170 cd/m$^2$ under 0 lx and 150 lx conditions, respectively [25], and 58.5 cd/m$^2$ increased to 84 cd/m$^2$ when 100 lx was increased to 300 lx [26]. For absolute darkness (0 lx), both the current findings (20.63–36.2 cd/m$^2$) and those of Li et al. (2013) (11 cd/m$^2$) [23] fell into the range identified by Na et al. (2015) (10–40 cd/m$^2$) [21]. However, Ye et al. (2014) [25] reported a much higher level of 55 cd/m$^2$. This may be due to the differential study approach employed in the study by Ye et al. (2014), in which participants were asked to adjust screen luminance subjectively to an optimal level according to their experience. Their findings also suggested that the self-adjusted optimal screen luminance levels were relatively higher than the levels automatically adjusted by the phone [25]. For the 100 lx illuminance level, the current findings were comparable to previous findings [23,26,28], whereas they were much lower than those of Kim et al. (2017) [27]. The property of a screen may be a contributor to this discrepancy. The highest luminance was 868 cd/m$^2$ in the study by Kim et al. (2017) [27], while 106.7 cd/m$^2$ was employed in the current study. In fact, in their follow-up study, they investigated the potential effects of the highest luminance level and found that the highest luminance levels played a key role and that the range of appropriate luminances varied according to the highest luminance levels [42]. More maximum levels of white luminance should be used in future studies during nighttime.

To the best of our knowledge, the current study was the first to explore the fitted curve of optimum screen luminance under ambient illuminance using data on subjective ratings of visual comfort, visual fatigue, screen brightness and satisfaction during nighttime. The findings also supported the necessity of including multiple indicators of subjective feelings when determining the optimum screen luminance under different ambient illumination conditions.

Some limitations should be noted in the current study. First, the current study only employed subjective ratings of visual comfort and visual fatigue; therefore, multiple objective indicators, including task performance and physiological activity (such as electroencephalogram (EEG) [56] and hemodynamic response [57]; see reviews [9,58]) that could be involved in LCD screen use are required in future studies. Second, the order of ambient illuminance was fixed and increased from 0 lx to 100 lx, as the speed of light adaption is faster than that of dark adaption in humans. Would the findings be different if the opposite order of ambient illuminance manipulation was employed? The potential influence of the order effect would be an interesting question and deserves more attention in the future. Third, several previous studies have also demonstrated that the preference of screen luminance and ambient illuminance may differ for young and elderly populations [13]. Since the current study explored only young people, further research is required to test whether the current findings on optimal screen luminance levels under different ambient illuminance conditions are age dependent. With the widespread use of smartphones and tablets, future work would be more mobile and pose new challenges regarding the determination of lighting parameters and luminance levels. A recent review showed that the use of smartphones and tablets would cause the same ocular and visual discomfort caused by computers [59]. Hence, whether the current findings could transfer to the use of other types of media with LCD screens (such as smartphones and tablets) remains largely unknown and warrants further investigation. Finally, the obtained results are representative of a very specific environmental condition and are probably not transferrable to other spaces characterized by other luminance distributions; thus, the generalizability of the present findings warrants further testing.

## 5. Conclusions

The current study fitted curves between screen luminance and ambient illuminance and obtained an optimum screen luminance curve under different ambient illuminance conditions in the evening. The results showed that optimum luminance of computer LCD screens increased with ambient illuminance, and the optimum ambient illuminance and screen luminance level for working in the evening were 13.08–62.16 lx and

20.63–75.15 cd/m$^2$, respectively. Therefore, ambient illuminance and screen luminance should be kept at a relatively low range to minimize the potentially detrimental effects of screen light and ambient light on biological functions at night.

**Supplementary Materials:** The following are available online at https://www.mdpi.com/article/10.3390/app11094108/s1, results of 5 (Ambient Illuminance: 0, 25, 50, 75, and 100 lx) × 6 (Screen Luminance: 3.87, 21.47, 42.74, 64.12, 84.77 and 106.7 cd/m$^2$) repeated analysis of variance (ANOVA). Figure S1: Subjective ratings of visual comfort with screen luminance under different ambient illuminance conditions. Figure S2: Subjective ratings of illuminance satisfaction with different screen luminance conditions. Figure S3: Subjective ratings of visual fatigue under ambient illuminance with different screen luminance conditions.

**Author Contributions:** Conceptualization, T.R.; Data curation, Y.Z. and H.S.; Formal analysis, Y.Z. and Q.-W.C.; Funding acquisition, G.Z.; Investigation, Y.Z. and H.S.; Project administration, Y.Z.; Software, Q.-W.C.; Supervision, T.R. and G.Z.; Validation, T.R.; Visualization, Q.-W.C.; Writing—original draft, Y.Z. and H.S.; Writing—review and editing, Q.-W.C. and T.R. All authors have read and agreed to the published version of the manuscript.

**Funding:** This research was funded by the National Key Research and Development Program of China (No. 2016YFB0401202); the Program for Chang Jiang Scholars and Innovative Research Teams in Universities (No. IRT_17R40); Guangdong Provincial Key Laboratory of Optical Information Materials and Technology (Grant No. 2017B030301007); the Science and Technology Program of Guangzhou (No. 2019050001); and the National Center for International Research on Green Optoelectronics, MOE International Laboratory for Optical Information Technologies, and The 111 Project.

**Institutional Review Board Statement:** The study was conducted according to the guidelines of the Declaration of Helsinki, and approved by the Institutional Review Board of South China Normal University.

**Informed Consent Statement:** Informed consent was obtained from all subjects involved in the study.

**Data Availability Statement:** The data used to support the findings of this study are available from the corresponding author upon reasonable request.

**Conflicts of Interest:** The authors declare no conflict of interest. The funders had no role in the design of the study; in the collection, analyses, or interpretation of the data; in the writing of the manuscript; or in the decision to publish the results.

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
