# Peer review of "Investigation of the Optimum Display Luminance of an LCD Screen under Different Ambient Illuminances in the Evening"

_applsci, doi:10.3390/app11094108_

Round 1

Reviewer 1 Report

Zhou et al present a paper in which they report subjective ratings of user comfort when reading from a computer monitor with at different display luminance and ambient illumination levels. I found the topic interesting and for the most part the paper well written. I did find some difficulties parsing the results, and I have some concerns regarding the analyses performed and reported. I provide several suggestions on how to better the presentation of the results. I also strongly suggest the authors to make their data and analyses publicly available in a public data repository. I provide detailed comments in the attached pdf.

Author Response

Dear editor in-chief and reviewers,

Thank you for the opportunity to revise our manuscript. We appreciate the constructive comments and suggestions. These were of great value for improving the quality of our article. Based on the comments, careful modifications have been made to our previous version. In the current version, all changes were highlighted within the document by blue colored text. After this revision, we also wrote a point-by-point response letter to the reviewers’ questions/comments.

We hope the revised manuscript will be acceptable for publication in Applied Sciences. Thank you very much for your time.

Reviewer 2 Report

The paper entitled “Investigation of the optimum display luminance of an LCD screen under different ambient illuminances in the evening” presents results of experiments aiming at identifying the optimal lighting conditions of a test environment (described by means of the illuminance at the eye level) to read on a LCD screen, the luminance of which was set according to six different levels. Optimal conditions were identified thanks to tests performed by 33 people, containing questions about screen brightness, visual comfort, visual fatigue and satisfaction about illuminance levels. The topic is interesting, worth being investigated, and the text is well-written, however, in my opinion the paper suffers of some methodological lacks that question the reliability of the results.

First of all, the lighting conditions analyzed in the experiment are very specific and the obtained results seem not exportable to other contexts. When evaluating discomfort conditions due to the use of computer screens, it is fundamental to consider the luminance contrast between the screen and the background and the luminance distribution in the visual field. According to the description of the test-room, the experiment was performed in a workstation surrounded by partitions, separating it from the three others and probably creating a restricted space. Moreover, it seems that the only light source in the room is a desk LED lamp for each workstation. Generally, these types of devices determine proper light illuminance at the task area, but creates great contrasts with the surroundings, leaving the rest of the room in darker conditions. Giving these two factors (the characteristics of the space and the selected luminaires), the obtained results are representative of a very specific environmental conditions and probably are not exportable to other spaces characterized by other luminance distributions. 

Except for these considerations, in my opinion the factor most affecting the reliability of the results is the way the test is administered. In about 40 minutes each subject evaluates 30 light scenes (5 illuminance levels per 6 luminance values) and he/she is exposed to each of them for only 30 seconds, before answer to the related questions. Time is not enough to well adapt to the different lighting conditions and the sequence of administration is too stressful for the subjects. This can invalidate the results. Moreover, considering that the goal is to evaluate comfort conditions and visual fatigue, each session should be last more time since discomfort sensations can be not so immediate, but can manifest in longer time. 

Author Response

(The authors gave the same response as above.)

Round 2

Reviewer 1 Report

I appreciate the author’s efforts in addressing my previous comments. The manuscript is much clearer now. However, I now that more details are provided, I have some concerns regarding the fitting procedures. Specifically, I think the authors might not be testing for overfitting correctly. Additionally, I notice the authors fit the average participant ratings, which means they are likely fitting high-order polynomials to noise. I think it would be more correct to fit their data individually for each participant, and then report the average function fits. I provide more details in the attached document. If the authors can fix these final issues with the analyses I would be inclined to endorse the publication of this work.

Author Response

Thank you for the opportunity to revise our manuscript. We appreciate the constructive comments and suggestions. These were of great value for improving the quality of our article. Based on the comments, careful modifications have been made to our previous version. In the current version, all changes were highlighted within the document by blue colored text. After this revision, we also wrote a point-by-point response letter to the reviewers’ questions/comments.

We hope the revised manuscript will be acceptable for publication in Applied Science. Thank you very much for your time.

Round 3

Reviewer 1 Report

I apologize to the authors for taking so long to look over their revisions. I am satisfied with how they have addressed my concerns and have no further comments.